

# Boundary curves of individual items in the distribution of total depressive symptom scores approximate an exponential pattern in a general population

Shinichiro Tomitaka[1,2], Yohei Kawasaki[2], Kazuki Ide[2], Maiko Akutagawa[2], Hiroshi Yamada[2], Toshiaki A. Furukawa[3] and Yutaka Ono[4]

[1] Department of Mental Health, Panasonic Health Center, Tokyo, Japan
[2] Department of Drug Evaluation and Informatics, Graduate School of Pharmaceutical Sciences, University of Shizuoka, Shizuoka, Japan
[3] Department of Health Promotion and Human Behavior, Department of Clinical Epidemiology/Graduate School of Medicine/School of Public Health, Kyoto University, Kyoto, Japan
[4] Center for the Development of Cognitive Behavior Therapy Training, Tokyo, Japan

Corresponding author
Shinichiro Tomitaka,
tomitaka.shinichiro@jp.
panasonic.com

## ABSTRACT

**Background:** Previously, we proposed a model for ordinal scale scoring in which individual thresholds for each item constitute a distribution by each item. This lead us to hypothesize that the boundary curves of each depressive symptom score in the distribution of total depressive symptom scores follow a common mathematical model, which is expressed as the product of the frequency of the total depressive symptom scores and the probability of the cumulative distribution function of each item threshold. To verify this hypothesis, we investigated the boundary curves of the distribution of total depressive symptom scores in a general population.

**Methods:** Data collected from 21,040 subjects who had completed the Center for Epidemiologic Studies Depression Scale (CES-D) questionnaire as part of a national Japanese survey were analyzed. The CES-D consists of 20 items (16 negative items and four positive items). The boundary curves of adjacent item scores in the distribution of total depressive symptom scores for the 16 negative items were analyzed using log-normal scales and curve fitting.

**Results:** The boundary curves of adjacent item scores for a given symptom approximated a common linear pattern on a log normal scale. Curve fitting showed that an exponential fit had a markedly higher coefficient of determination than either linear or quadratic fits. With negative affect items, the gap between the total score curve and boundary curve continuously increased with increasing total depressive symptom scores on a log-normal scale, whereas the boundary curves of positive affect items, which are not considered manifest variables of the latent trait, did not exhibit such increases in this gap.

**Discussion:** The results of the present study support the hypothesis that the boundary curves of each depressive symptom score in the distribution of total depressive symptom scores commonly follow the predicted mathematical model, which was verified to approximate an exponential mathematical pattern.

## INTRODUCTION

Depression is a major public health concern as one of the leading causes of disease burden worldwide, with an estimated 350 million people of all ages affected around the globe (*Moussavi et al., 2007*). Given that the presence of depressive symptoms is closely linked with clinical levels of depression, there has been great interest in understanding the distribution of depressive symptoms in the general population (*Blazer et al., 1994*; *Kroenke et al., 2009*).

Building a model or formulating a theory based on empirical data is fundamental to advancing scientific practice. However, despite the accumulation of knowledge regarding depressive symptoms, little information is available for the development of a mathematical model that may predict the distribution of depressive symptoms.

In developing a model of the distribution of depressive symptoms in the general population, it is first necessary to identify reproducible phenomenon. Several recent studies with large sample sizes have revealed that total depressive symptom scores in the general population approximate an exponential pattern, except at the lowest end of the range of scores. In an analysis of data from nearly 10,000 non-psychotic respondents to the British National Household Psychiatric Morbidity Survey, (*Adult Psychiatric Morbidity in England, 2007*), *Melzer et al. (2002)* observed that an exponential curve provided the best fit for total depressive and neurotic symptom scores on the Revised Clinical Interview Schedule (CIS-R) (*Lewis et al., 1992*). The authors of the present study have similarly observed that the right tail of the distribution of total depressive symptom scores on the Center for Epidemiologic Studies Depression Scale (CES-D) follows an exponential curve, based on data from a national survey of the Japanese population including data from nearly 25,000 respondents (*Tomitaka, Kawasaki & Furukawa, 2015a*). A similar study involving a large sample of Japanese employees further supported the exponential pattern of CES-D scores, except at the lowest end of the scale (*Tomitaka et al., 2016b*). We considered the aforementioned phenomenon in the development of our mathematical model of the distribution of depressive symptoms and conducted research regarding the mechanisms responsible for the generation of an exponential distribution as follows.

The CES-D allows individual to self-rate the frequency of a variety of depressive symptoms (sadness, fatigue, etc.) on a scale consisting of four possible responses: "rarely (less than 1 day)," "some (1–2 days)," "occasionally (3–4 days)," and "most of the time (5–7 days)" (*Radloff, 1977*). Recently, we have demonstrated that, in a general population, responses to each of the 16 individual items related to negative symptoms of depression on the CES-D tend to exhibit exponential patterns between the "some" and "most" responses, while this pattern is not observed for "rarely" responses. (*Tomitaka, Kawasaki & Furukawa, 2015b*). Based on this finding and those of previous reports, we then proposed that the 16 items related to negative depressive symptoms are manifest variables influenced by a unidimensional latent trait of depressive symptoms, which itself

follows an exponential distribution (*Tomitaka, Kawasaki & Furukawa, 2015b*; *Tomitaka et al., 2016a*).

In order to explain the aforementioned findings that total depressive symptom scores approximate an exponential pattern in the general population, we proposed a model of ordinal scales for unidimensional latent traits (*Tomitaka et al., in press*). This model assumes that individual thresholds for each depressive symptom differ from each other, with each item exhibiting its own unique distribution. When the degree of a participant's latent trait is greater than that of the threshold for the specific depressive symptom, the specific symptom is expected to be present. In this simulation study, we assumed that the threshold for each depressive symptom forms a normal distribution and set the simulated threshold distribution accordingly. The simulation study confirmed that total scores of ordinal scale items approximate a pattern similar to that of the latent trait distribution, suggesting that total scores along an ordinal scale correspond to the latent trait (*Tomitaka et al., in press*).

In general, the most important consideration when evaluating a mathematical model is whether the model is consistent with empirical data. According to our model of ordinal scales, the total score of depressive symptoms, which correspond to the latent trait, is expressed as an exponential distribution, and the distribution of the threshold for each depressive symptom is expressed as a normal distribution. Since the probability of each score for any given item depends on whether the item threshold is smaller than the degree of the individual's latent trait, the cumulative distribution function of each item threshold corresponds to the probability of each item score. Thus, the boundary curve of each item score's frequency for a given degree of total depressive symptom scores can hypothetically be expressed as the product of the frequency of the total depressive symptom scores and the probability of the cumulative distribution function of each item threshold. However, little research has been conducted regarding such a distribution. Thus, in the present study, we investigated the whether the boundary curves, which represent the boundaries of each item score's frequency for a given degree of total depressive symptom scores, followed a common mathematical pattern predicted by the present model. This study used data from a large, cross-sectional national survey of the general Japanese population (*Ministry of Health, Labor and Welfare, Statistics and Information Department, 2002*).

In the present study, we first investigated the boundary curves of adjacent item scores in the distribution of total depressive symptom scores for the 16 negative items. After confirming that the boundary curves of adjacent item scores of each depressive symptom item commonly approximated an exponential pattern, the boundary curves associated with the absence of multiple depressive symptoms were analyzed to determine whether they approximated an exponential pattern. Finally, to ascertain whether the common mathematical pattern of the boundary curves is specific to the 16 negative affect items following the distribution of the unidimensional latent trait of depressive symptoms, we examined whether the boundary curve of four positive affect items, which do not follow this latent trait distribution, follow the same pattern observed for 16 negative affect items.

## MATERIALS AND METHODS

The present study utilized data from the Active Survey of Health and Welfare (ASHW) conducted by the Japanese Ministry of Health, Labor, and Welfare in 2000 (*Ministry of Health, Labor and Welfare, Statistics and Information Department, 2002*). The ASHW is an annual nationwide survey conducted by the Japanese Government to collect data necessary for policy making and health promotion in compliance with the Statistics Act. Legal and ethical approval for the ASWH was provided by the Japanese Ministry of Health, Labor, and Welfare. In 2000, the ASHW examined depressive symptoms among a representative sample from the general Japanese population. To ensure that the sample was adequately representative, survey participants were selected from individuals 12 years and older living across 300 communities in Japan. These communities were selected from 881,851 precincts identified in the 1995 Census using a stratified sampling design. The survey was conducted anonymously, and verbal informed consent was obtained from all participants and legal guardians. The data and methods used by the survey have been described in detail in a previous report (*Ministry of Health, Labor and Welfare, Statistics and Information Department, 2002*).

The questionnaire was returned by 32,729 respondents, though the response rate was not published by the Ministry of Health, Labor, and Welfare and Health. However, the response rates for similar surveys conducted three and four years earlier were 87.1 and 89.6%, respectively (*Kaji et al., 2010*). Therefore, we assumed that the response rate for the ASWH survey to be greater than 80%.

The Japanese Ministry of Health, Labor, and Welfare examined our study and permitted us to perform a secondary analysis of the anonymized data from the ASWH in compliance with the Statistics Act. The present study was approved in 2014 by the ethics committee of the Panasonic Health Center (approval number, 2014-1). The authors assert that all procedures contributing to this work were compliant with the ethical standards of the relevant national and institutional committees on human experimentation, and were in accordance with the Helsinki Declaration of 1975 as revised in 2008.

We excluded 1,394 respondents owing to the suspect validity of their responses (i.e., those who answered "rarely" or "most" for all items, regardless of the nature of the item). A total of 9,588 respondents were also excluded from the sample owing to missing information on one or more key study variables (i.e., depressive symptoms, age, sex). The final sample consisted of 21,040 respondents between 12 and 98 years of age (ages 12–19; N = 2,457 (male; n = 1,269), ages 20–29; N = 3,748 (male; n = 1,788), ages 30–39; N = 3,761 (male; n = 1,783), ages 40–49; N = 3,629 (male; n = 1,788), ages 50–59; N = 3,569 (male; n = 1,800), ages 60–69; N = 2,253 (male; n = 1,155), ages 70–79; N = 1,161 (male; n = 517), ages 80–89; N = 412 (male; n = 108), ages 90–98; N = 50 (male; n = 15)).

### Measures

Depressive symptoms were assessed using the Japanese version of the CES-D (*Shima et al., 1985*). This 20-item scale assesses the frequency of a variety of depressive symptoms

experienced within the previous week (0 = rarely or none of the time (less than 1 day), 1 = some or little of the time (1–2 days), 2 = occasionally or a moderate amount of time (3–4 days), and 3 = most or all of the time (5–7 days)), yielding a total score of 0–60 (*Radloff, 1977*). Higher scores indicate greater psychological distress. The 20 items of the CES-D are generally grouped into the following four subscales: depressive mood (items 3, 6, 9, 10, 14, 17, and 18); somatic symptoms (items 1, 2, 5, 7, 11, 13, and 20); interpersonal relations (items 15 and 19); and positive affect (items 4, 8, 12, and 16). The positive affect items are reverse-scored. In our previous work, we revealed that the 16 items related to depressive mood, somatic symptoms, and interpersonal relations follow a common mathematical model, while the four items related to positive affect do not, suggesting that these items/symptoms associated with positive affect are not manifest variables of the unidimensional latent trait (*Tomitaka, Kawasaki & Furukawa, 2015b*).

The boundary curves of adjacent item scores represent the boundaries of the frequencies for response categories, including "rarely," "some," "occasionally," and "most." The boundary curves of adjacent item scores in the distribution of total depressive symptom scores were calculated according to the frequency of each item score and analyzed using a log-normal scale. The fitting curve for exponential, linear, and quadratic models were estimated using the least square method. JMP Version 11 for Windows (SAS Institute, Inc., Cary, NC, USA) was used to calculate descriptive statistics and frequency distribution curves.

## RESULTS

### Boundary curves of adjacent item scores of individual depressive symptoms

Figure 1 depicts the distribution of total scores for 16 negative items (green lines) and the boundary curves (magenta line, yellow line, and blue line), which divide the distribution of total depressive symptom scores by the adjacent scores of each depressive symptom. The magenta, yellow, and blue lines represent the boundaries between score 0 and score 1, score 1 and score 2, and score 2 and score 3, respectively.

On a normal scale, the boundary curves of adjacent scores for item 1 (Fig. 1A), item 2 (Fig. 1B), and item 3 (Fig. 1C) exhibited a right skewed distribution (Figs. 1A–1C), while they exhibited a linear pattern on a log-normal scale (Figs. 1D–1F). The gaps between the total score curve and boundary curves of adjacent scores widened with increasing total depressive symptom scores (Figs. 1D–1F). Analysis of the boundary curves of the remaining 13 negative depressive symptom items (Figs. S1–S5) revealed the same mathematical pattern as that shown in Fig. 1.

Table 1 outlines the empirical constants and coefficient of determination $R^2$ values for linear, quadratic, and exponential fits, analyzed with respect to the boundary curves of adjacent scores for item 1, item 2, and item 3. Results for the remaining 13 negative depressive symptoms are presented in Table S1. In all 48 boundary curves, the coefficients of determination for the exponential fit were higher than those for linear or quadratic fits. The median coefficients of determination for the linear, quadratic, and exponential

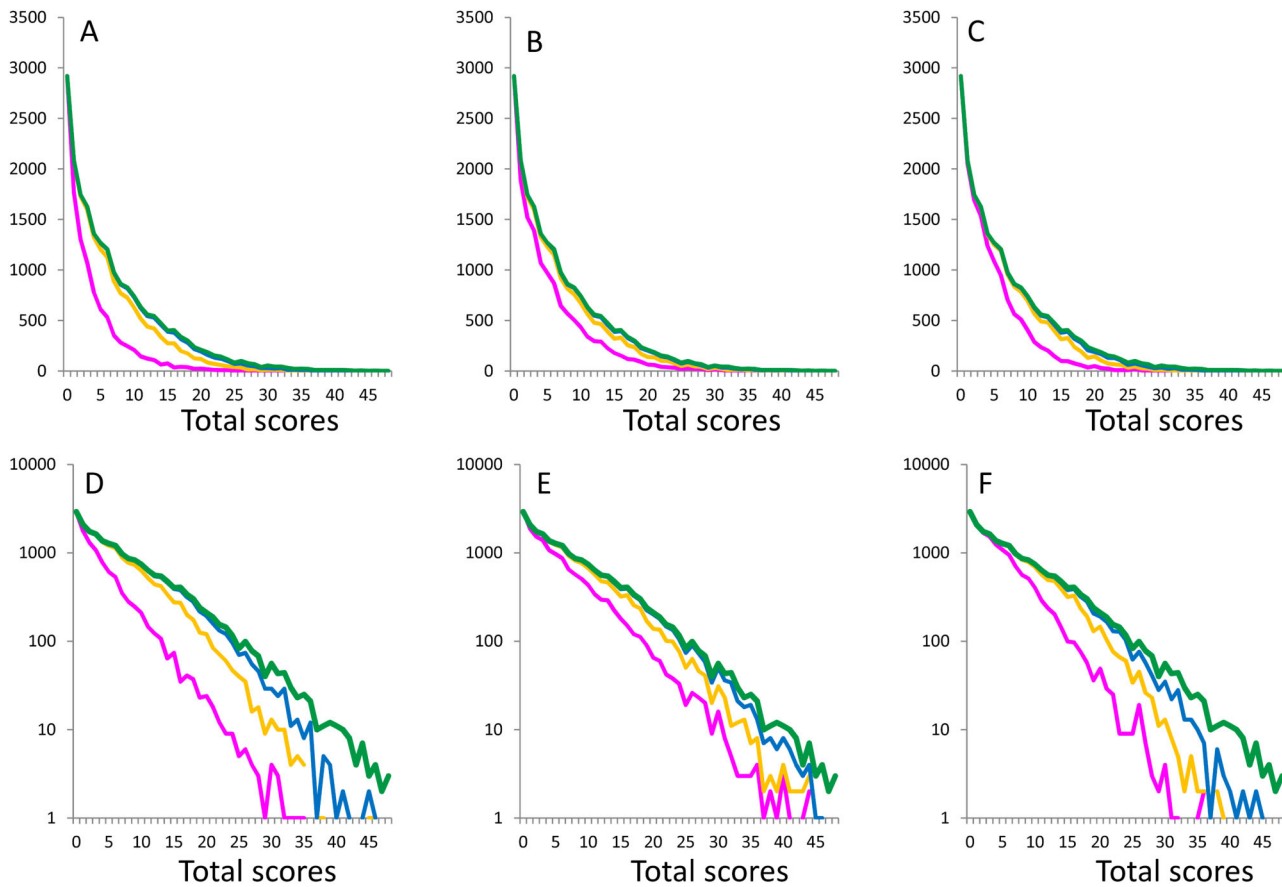

**Figure 1 Distributions of the total scores of 16 items (green lines) and the boundary curves of the adjacent score for each item.** (A, D) Boundary curves of item 1, (B, E) item 2 and (C, F) item 3 with a normal scale and a log-normal scale are presented, respectively. Magenta, yellow, and blue lines represent the boundary curves between score 0 and score 1, score 1 and score 2, and score 2 and score 3 of Likert scale scores (0-1-2-3), respectively.

fits were 0.61, 0.91, and 0.97, respectively, suggesting that an exponential distribution best fit the observed boundary curve data.

## Boundary curves in the absence of multiple symptoms

Boundary curves in the absence of symptoms related to two items (item 1 and item 2, Figs. 2A and 2D), four items (item 1, item 2, item 3, and item 5; Figs. 2B and 2E), and six items (item 1, item 2, item 3, item 5, item 6, and item 7; Figs. 2C and 2F) were considered representative of the absence of multiple symptoms. On a normal scale, the boundary curves in the absence of multiple symptoms exhibited a right-skewed distribution (Figs. 2A–2C), while they exhibited a linear pattern on a log-normal scale (Figs. 2D–2F). As indicated by the double-headed arrows, the gap between the total score curve and boundary curve widened as total depressive symptom scores increased.

Table 2 outlines the empirical constants and coefficient of determination $R^2$ values for linear, quadratic, and exponential fits, analyzed with respect to the boundary curves in the absence of multiple symptoms. Curves of fit using the exponential model and the observed data in Figs. 2A–2C yielded coefficients of determination of 0.99, 0.97, and 0.93,

**Table 1 Coefficient of determination $R^2$ values for the fit of linear, quadratic, and exponential curves to boundary curve data for 16 depressive symptoms.** Empirical constants are given for linear fitting with $Y = a * X + b$, for quadratic fitting with $Y = a * X^2 + b * X + c$, for exponential fitting with $Y = a * e^{b * X}$. The coefficient of determination $R^2$ values for exponential fits were higher than those for linear or quadratic fits for all boundary curves.

| Boundary curve | Linear | | | Quadratic | | | | Exponential | | |
|---|---|---|---|---|---|---|---|---|---|---|
| | a | b | $R^2$ | a | b | c | $R^2$ | a | b | $R^2$ |
| Item 1, score 0–1 | −22.7 | 789 | 0.37 | 1.7 | −106 | 1,496 | 0.68 | 2,482 | −0.23 | 0.99 |
| Item 1, score 1–2 | −34.1 | 1,227 | 0.59 | 1.9 | −131 | 2,051 | 0.89 | 4,369 | −0.19 | 0.98 |
| Item 1, score 2–3 | −56.7 | 1,615 | 0.75 | 3.1 | −171 | 2,339 | 0.94 | 5,172 | −0.18 | 0.95 |
| Item 2, score 0–1 | −28.9 | 1,029 | 0.50 | 1.8 | −119 | 1,796 | 0.82 | 2,793 | −0.19 | 0.98 |
| Item 2, score 1–2 | −34.8 | 1,263 | 0.61 | 1.9 | −129 | 2,060 | 0.90 | 4,321 | −0.17 | 0.98 |
| Item 2, score 2–3 | −35.9 | 1,320 | 0.65 | 1.8 | −127 | 2,091 | 0.91 | 4,274 | −0.16 | 0.98 |
| Item 3, score 0–1 | −30.3 | 1,064 | 0.49 | 2.0 | −130 | 1,914 | 0.83 | 4,782 | −0.24 | 0.98 |
| Item 3, score 1–2 | −35.2 | 1,273 | 0.61 | 1.9 | −132 | 2,094 | 0.91 | 5,366 | −0.20 | 0.96 |
| Item 3, score 2–3 | −36.0 | 1,316 | 0.65 | 1.8 | −128 | 2,097 | 0.91 | 5,320 | −0.18 | 0.96 |

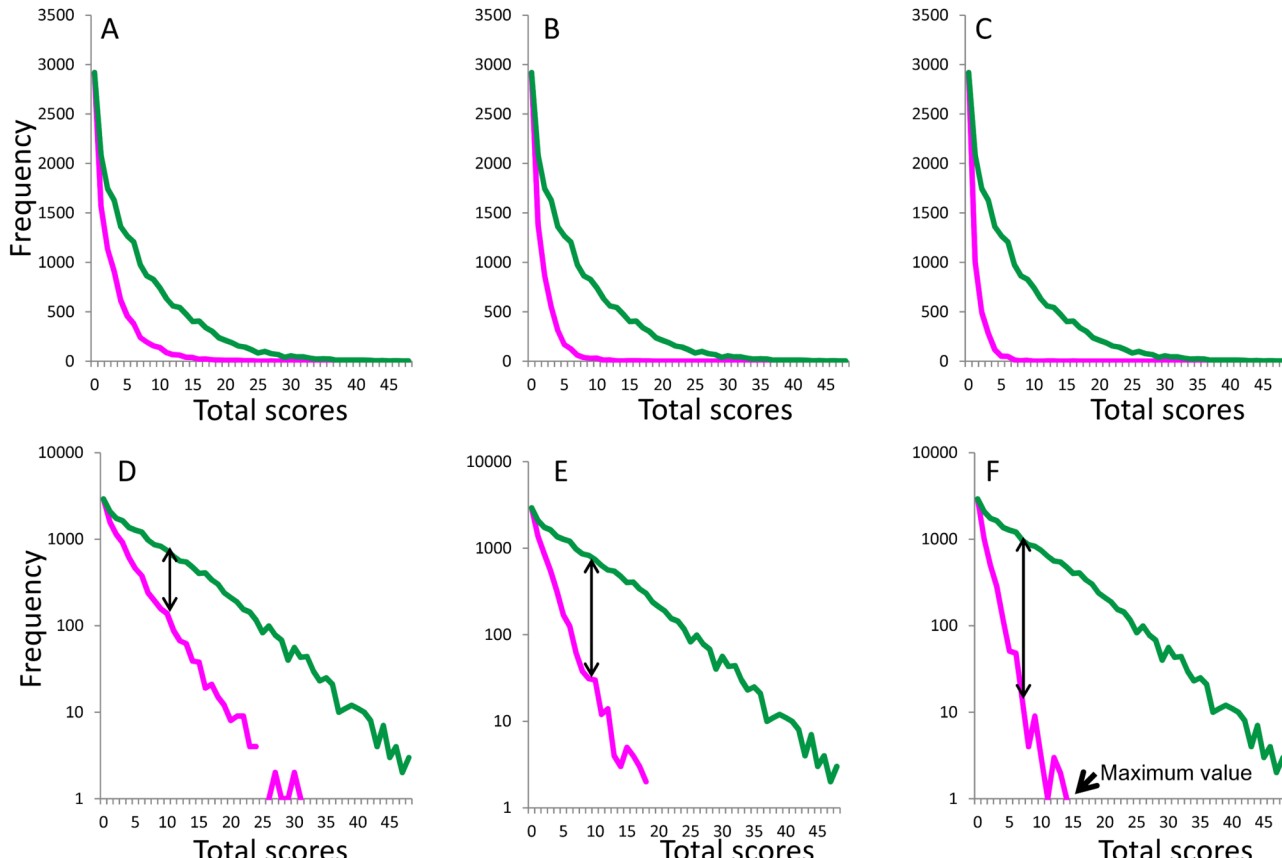

**Figure 2 Boundary curves (magenta lines) in absence of two, four, and six items.** Boundary curves of two items (item 1 and item 2; A, D), four items (item 1, item 2, item 3, and item 5; B, E), and six items (item 1, item 2, item 3, item 5, item 6, and item 7; C, F) with normal scales and log-normal scales are presented, respectively.

**Table 2 Coefficient of determination $R^2$ values for the fit of linear, quadratic, and exponential curves to boundary curve data in the absence of multiple symptoms.** Empirical constants are given for linear fitting with $Y = a * X + b$, quadratic fitting with $Y = a * X^2 + bX + c$, and exponential fitting with $Y = a * e^{b * X}$. The coefficient of determination $R^2$ values for exponential fits were higher than those for linear or quadratic fits for boundary curves in the absence of multiple symptoms.

| Boundary curve | Linear | | | Quadratic | | | | Exponential | | |
|---|---|---|---|---|---|---|---|---|---|---|
| | a | b | $R^2$ | a | b | c | $R^2$ | a | b | $R^2$ |
| 2 items | −19.7 | 679 | 0.31 | 1.5 | −96 | 1,329 | 0.61 | 2,410 | −0.26 | 0.99 |
| 4 items | −15.0 | 508 | 0.21 | 1.3 | −80 | 1,062 | 0.45 | 2,320 | −0.40 | 0.97 |
| 6 items | −11.7 | 394 | 0.14 | 1.1 | −65 | 849 | 0.33 | 2,317 | −0.57 | 0.94 |

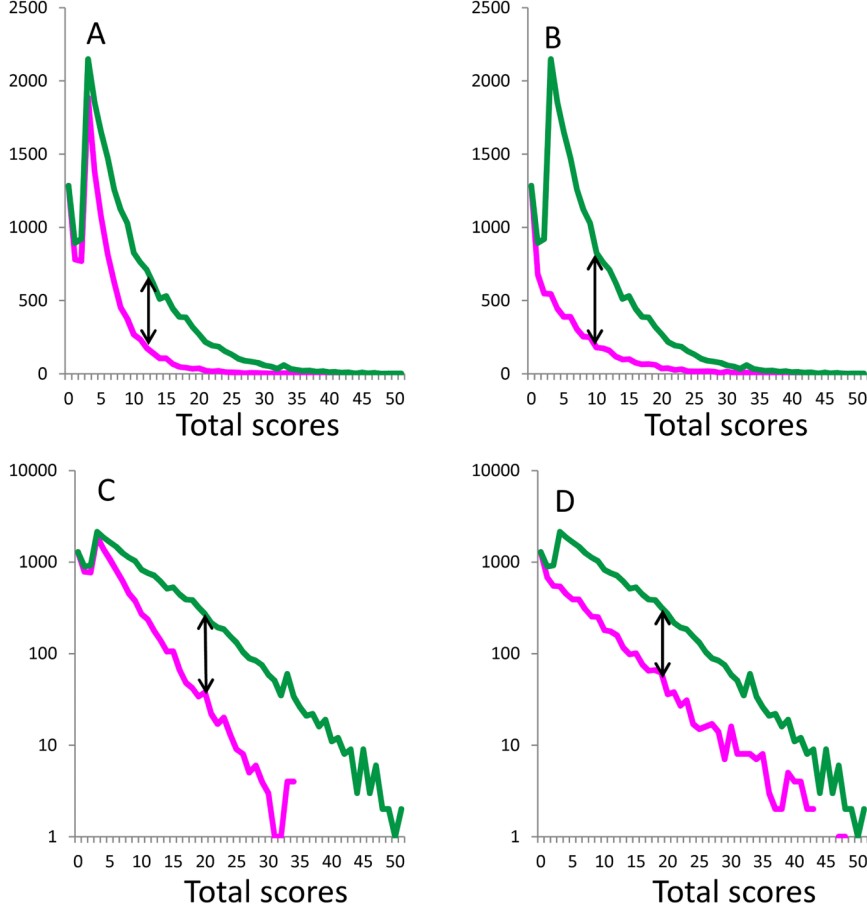

**Figure 3 Boundary curves (magenta lines) of a negative depressive symptom item and a positive affect symptom item in the distribution of total depressive symptom scores for 17 depressive symptom items.** (A, C) Boundary curves of item 1 and (B, D) item 4 with a normal scale and a log-normal scale are presented, respectively.

respectively, suggesting that exponential distributions had the best fit with the observed data.

The rate parameter of the curve fit for the boundary curve (constant b) decreased as the number of symptoms absent increased (Table 2), consistent with the finding that the slope

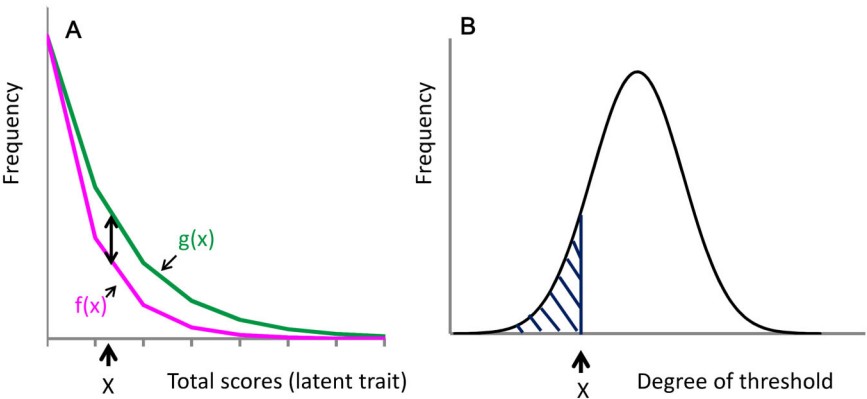

**Figure 4 Distribution of the total depressive symptom scores of 16 negative items, boundary curves in the absence or presence of each symptom item (magenta line) and the distribution of each item threshold.** (A) The probability of a specific symptom is expressed as the ratio of the frequency of the presence of a specific item (black double-headed arrow) to the frequency of total depressive symptoms (green line). (B) The probability of a specific symptom with the total score of X also corresponded to the cumulative distribution function of each item threshold (shaded area).

of the boundary curve on a log-normal scale became steeper as the number of symptoms absent increased. Moreover, the range of total scores narrowed as the number of symptoms absent increased (Figs. 2D–2F). The data presented in Fig. 2F indicates that, in the absence of symptoms related to the six abovementioned items, the maximum range of total scores spanned 16 points.

## Boundary curves of positive affect items in the distribution of total depressive symptom scores for 17 depressive symptom items

To ascertain whether the common mathematical pattern of the boundary curves is specific to the 16 depressive symptom items, which are the manifest variables of a unidimensional latent trait, we compared a representative curve (item 1) to the boundary curve of an item related to positive affect (item 4), which is not a manifest variable of unidimensional latent trait. While the boundary curves of both item 1 and item 4 exhibited a common right-skewed distribution (Figs. 3A and 3B) on a normal scale, the gap between the total score curve and boundary curves differed between the two (Figs. 3A and 3B). For item 1, the gap between the total score and boundary curves exhibited a pattern similar to that observed in Figs. 1 and 2, increasing to a certain point and then decreasing to zero at the end of the range (Fig. 3A). For item 4, however, this gap decreased until the end of the range, beginning with scores of 3 (Fig. 3B).

While the boundary curves of both item 1 (Fig. 3C) and item 4 (Fig. 3D) exhibited a linear pattern in the rage of total scores from 3–51 points on a log-normal scale, the gap between the total score curve and boundary curve again exhibited different patterns between items 1 and 4 (Figs. 3C and 3D). For item 1, the gap between the total score curve and boundary curve widened as total depressive symptom scores increased, similar to the pattern observed in Figs. 1D–1F and 2D–2F (Fig. 3C). Conversely, for item 4, the gap between the total score curve and boundary curve remained stable (Fig. 4D).

The curves of fit according to an exponential model were calculated for data from the total score curve in Fig. 3C (y = 1823e$^{-0.23x}$, $R^2$ = 0.97), the boundary curve in Fig. 3C (y = 3327e$^{-0.15x}$, $R^2$ = 0.98), the total score curve in Fig. 3D (y = 3327e$^{-0.15x}$, $R^2$ = 0.98), and the boundary curve in Fig. 3D (y = 580e$^{-0.14x}$, $R^2$ = 0.97). Consistent with log-normal scale findings, exponential curve fitting showed that, while exponential distributions fitted the boundary curves of both item 1 and item 4, the rate parameters of the curves of fit for the total score curve and boundary curve were similar for item 4 (−0.15 vs. −0.14) but not for item 1 (−0.23 vs. −0.15). Analysis of the boundary curves of the remaining three items associated with positive affect (Fig. S6) revealed the same mathematical pattern as that shown in Figs. 3B and 3D, suggesting that an increasing gap between the total score and boundary curves on a log-normal scale is specific to the 16 depressive symptom items, which comprise the manifest variables of a unidimensional latent trait.

## DISCUSSION

In the present study, we aimed to verify whether boundary curves, which represent the boundaries of each item score for a given degree of total depressive symptom scores, followed a common mathematical pattern predicted by the present model. The main findings of this study are as follows: (1) regardless of item choice, the boundary curves for items associated with negative depressive symptoms approximated an exponential pattern; (2) on a log-normal scale, the gap between the total score and boundary curves for negative affect items, but not for positive affect items, increased as total depressive symptom scores increased; (3) the boundary curves associated with the absence of multiple symptoms commonly approximated an exponential pattern.

### Boundary curves for items related to negative depressive symptoms follow an exponential pattern

Regardless of item choice, the boundary curves for items associated with negative depressive symptoms exhibited a common linear pattern on a log-normal scale (Figs. 1–3). In addition, exponential fits showed markedly higher coefficients of determination than either linear or quadratic fits, suggesting that the boundary curves of negative affect items follow an exponential pattern.

Although we hypothesized that the boundary curve of each item score could be expressed as the product of the frequency of the total depressive symptom scores and the probability of the cumulative distribution function of each item threshold, we did not predict that the boundary curves for items associated with negative depressive symptoms approximated an exponential distribution. Mathematical speculations can be made about the conditions that enabled such an exponential pattern. As depicted in Fig. 4A, the frequency of the total depressive symptom scores for the 16 items associated with negative depressive symptoms may be expressed as an exponential distribution, g($x$) (green line), as follows for a total score of $x$:

$$g(x) = Ce^{-ax} \quad a > 0, \tag{1}$$

where $C$ is the frequency of total scores for the 16 negative items at a score of zero and $a$ is the parameter of the exponential function for these total scores.

The frequency of boundary curves between the absence and presence of each symptoms item is then expressed as $f(x)$ (Fig. 4A, magenta line). The probability of the presence of a given symptom, $\mathbf{P}(x)$, is expressed as the ratio of the frequency of that symptom (black two-headed arrow) divided by the frequency of total depressive symptom scores (green line) as follows:

$$\mathbf{P}(x) = \{g(x) - f(x)\}/g(x). \tag{2}$$

Figure 4B depicts the distribution of each item threshold according to the latent trait. According to the present model, the probability of the presence of a given symptom corresponds to the cumulative distribution function of each item threshold where the degree of the threshold for the given item is greater than the latent trait value of $x$ (shaded area).

Assuming a normal distribution for each item threshold, the cumulative distribution function of each item threshold may be expressed using a logistic distribution (Bowling et al., 2009). The probability of the presence of a given symptom is then expressed as follows:

$$\mathbf{P}(x) = 1/(1 + e^{-(x-\mu)/s}), \tag{3}$$

where $\mu$ is the mean, $s$ is a parameter of logistic distribution, and $s > 0$.

Figure 5 depicts $g(x)$ and $f(x)$ on a logarithmic scale. When $g(x)$ is expressed in logarithmic terms,

$$\log g(x) = -ax + \log C. \tag{4}$$

(Fig. 5, green line)

According to Eqs. (1), (2), and (3),

$$f(x) = g(x)(1 - \mathbf{P}(x)) = \{Ce^{-(a+1/S)X+\mu/s}\}/(1 + e^{-(x-\mu)/s}). \tag{5}$$

When $f(x)$ is expressed in logarithmic terms,

$$\log f(x) = -(a + 1/s)x + \log C + u/s - \log(1 + e^{-(x-\mu)/s}), \tag{6}$$

(Fig. 5, magenta curve)
where $\log(1 + e^{-(x-\mu)/s})$ decreases to zero with increasing values of $x$, and $\log(1 + e^{-(x-\mu)/s}) > 0$.

When the former portion of $\log f(x)$ is expressed as $\log h(x)$,

$$\log h(x) = -(a + 1/s)x + \log C + u/s. \tag{7}$$

(Fig. 5, blue dotted line)

As $x$ increases, $\log f(x)$ (magenta curve) approaches $\log h(x)$ (blue dotted line) due to the decrease in the value of $\log(1 + e^{-(x-\mu)/s})$. As depicted in Fig. 5, the distance between the blue and red lines (black two-headed arrow) decreases as $x$ increases. If the parameter $s$ is large enough, the distance between the blue and red lines mildly decreases

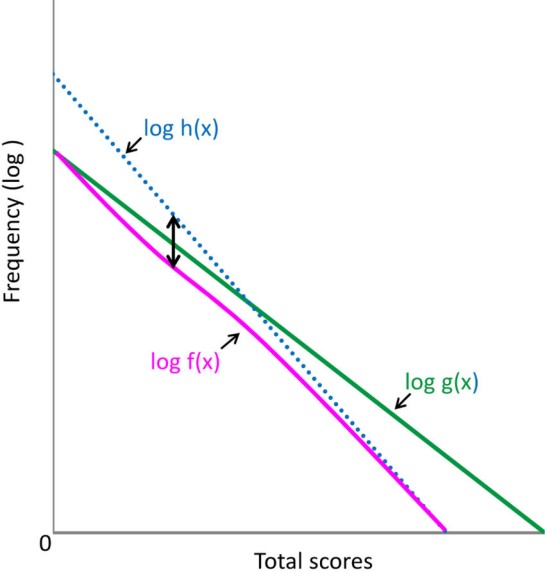

**Figure 5 Distribution of the total depressive symptom scores of 16 negative items (green line) and boundary curves in the absence or presence of each symptom item (magenta line) in a single logarithmic chart.** The blue dotted line is log h($x$). The distance between the blue line and magenta line (double arrow) decreased from $\log(1 + e^{-(x-\mu)/s})$ approached zero with $x$.

with increasing values of $x$. Thus, although log f($x$) is not a perfect linear function, it approximates a linear pattern.

## Increasing gap between the total score and boundary curves for negative symptom items on a log-normal scale

The results of the present study reveal that the gap between the total score curve and boundary curve for negative affect items increases as total depressive symptom scores increase on a log-normal scale (Fig. 1–3). In support of the finding, the estimated rate parameters of the exponential fit curves for the total score curve and boundary curve were not similar for negative affect items. The data expressed in Fig. 5 may help to explain these findings. As the total depressive symptom scores increase, log f($x$) (magenta line) distances itself from log g($x$) (green line) because log f($x$) approaches h($x$) (blue dotted line), resulting in the consistently increasing gap between the total score curve and boundary curves.

Conversely, for the item associated with positive affect, the gap between the total score curve and boundary curve did not increase as total depressive symptom scores increased on a log-normal scale (Fig. 3D). Consistent with this finding, the estimated rate parameters of the exponential fit curves for the total score curve and boundary curve were similar for positive affect items. Mathematically, the increasing gap corresponds to the cumulative distribution function of each item threshold, which has a continuous increasing property (Fig. 4). Thus, the increasing gap between the total score curve and boundary curve could be specific to negative affect items, which are considered to be manifest variables of a unidimensional latent trait. In general, it is difficult to differentiate
whether given items are manifest variables of the unidimensional latent trait, except when the latent trait follows an exponential distribution (*Tomitaka et al., 2016a*). Although most studies have utilized a taxometric analysis, which is a statistical technique specifically designed to determine whether a given construct is best conceptualized as discrete latent subcategories or as one continuous latent dimension (*Slade & Andrews, 2005*; *Okumura, Sakamoto & Ono, 2009*), the consistently increasing gap between the total score curve and boundary curve on a log normal scale may be useful in determining whether or not a given item is a manifest variable of a unidimensional latent trait. The finding that the gap between the total score curve and boundary curve for negative affect items increased as total depressive symptom scores increased on a log-normal scale, irrespective of the item subscale, supports a unidimensional latent trait of depressive symptoms.

## Boundary curves in the absence of multiple symptoms

The results of the present study indicate that the boundary curves associated with the absence of multiple symptoms commonly approximate an exponential pattern. In addition, as the number of symptoms absent increases, the slope of the boundary curves decreases on a log-normal scale (Figs. 2D–2F). Theoretically, the boundary curves associated with the absence of multiple symptoms depend on whether the largest item threshold is smaller than the degree of latent trait. If the largest item threshold follows a normal distribution, the boundary curves in the absence of multiple symptom items could approximate an exponential distribution.

From a clinical standpoint, the finding that the range of total depressive symptom scores is determined by the number of symptoms absent is noteworthy. In the present study, in the absence of six depressive symptoms, the maximum total score was 16 points, which is nearly equivalent to the threshold score for major depression in a Japanese population (*Shima et al., 1985*; *Wada et al., 2007*). These findings may be useful for the screening of clinical depression in a general population. Further studies are necessary to establish the relationship between the numbers of symptoms absent and total depressive symptom scores.

## Strengths and limitations

The present study had some limitations. First, participants did not undergo the standard psychiatric interview and diagnosis associated with a structured interview. Second, although we evaluated whether an exponential model provided a better fit than a linear or quadratic model, we did not examine the fits of other mathematical models. To the best of our knowledge, no other mathematical models have been proposed for the boundary curves of adjacent item scores other than the exponential model. Future studies can evaluate the comparative fit of other models to our empirical data as reported in our Supplemental Information. Third, in order to explain the mathematical pattern of boundary curves, we assumed a normal distribution for each item threshold. However, it is not clear whether a normal distribution is most appropriate for the

empirical data associated with each item threshold, indicating the need for further studies.

Conversely, there is a methodological advantage in the present investigation. The sample was representative of the general Japanese population, which reduced selection bias. In addition, the large sample size (N = 21,040) enabled us to elucidate patterns in the distributions of depressive symptom items. Finally, the present study provides important information regarding the boundary curves in the distribution of total depressive symptom scores. To the best of our knowledge, mathematical modeling remains poorly developed in the field of psychiatry. While a given individual's depressive symptoms are difficult to predict, large populations may follow a certain mathematical patterns. The present study thus proposes a statistical model for the relationship between the prevalence of each depressive symptom and total depressive symptom scores. The degree to which these results generalize to other ordinal scales, including other depression rating scales, intelligence tests, and personality questionnaires, requires further examination.

## CONCLUSIONS

The results of the present study support the hypothesis that the boundary curves of for scores associated with a given depressive symptom in the distribution of total depressive symptom scores exhibit a common mathematical pattern, raising the possibility that boundary curves for given items may be useful in evaluating whether the specific item is a manifest variable of the unidimensional latent trait.

## ACKNOWLEDGEMENTS

The authors would like to thank the Active Survey of Health and Welfare project for providing the data for this study.

### Funding
The authors received no funding for this work.

### Competing Interests
The authors declare that they have no competing interests.

### Author Contributions
- Shinichiro Tomitaka conceived and designed the experiments, performed the experiments, analyzed the data, wrote the paper, prepared figures and/or tables, reviewed drafts of the paper.
- Yohei Kawasaki reviewed drafts of the paper.
- Kazuki Ide reviewed drafts of the paper.
- Maiko Akutagawa reviewed drafts of the paper.
- Hiroshi Yamada reviewed drafts of the paper.
- Toshiaki A. Furukawa reviewed drafts of the paper.
- Yutaka Ono reviewed drafts of the paper.

## Human Ethics

The following information was supplied relating to ethical approvals (i.e., approving body and any reference numbers):

The present study was approved in 2014 by the ethics committee of Panasonic Health Center (approval number, 2014-1). The authors assert that all procedures contributing to this work comply with the ethical standards of the relevant national and institutional committees on human experimentation and with the Helsinki Declaration of 1975, as revised in 2008.

## Data Deposition

The raw data has been supplied as Supplemental Dataset Files.

## Supplemental Information

Supplemental information for this article can be found online at http://dx.doi.org/10.7717/peerj.2566#supplemental-information.

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
