# Peer review of "Boundary curves of individual items in the distribution of total depressive symptom scores approximate an exponential pattern in a general population"

_PeerJ, doi:10.7717/peerj.2566_

## Round 0.1 · original submission · Major Revisions

Thank you for your interesting paper. The reviewers have raised several issues, which I concur need addressing and the manuscript would benefit significantly from the following being addressed.

A deeper statistical analysis and justification for results would be helpful. In particular comparisons to other models that support the justification for this particular model. Directly addressing quantitatively, statistically, the model fit rather than qualitatively doing so would strengthen the paper.

Therefore, a major revision is required with clarification and support of the model and comparisons to other models is needed.

Reviewer 1 ·

Basic reporting

The article adheres to these standards.
Melzer reference is missing.

Experimental design

There should be more done to define and describe the role of boundary curve lines and their relevance to the current project. For example-how are the curves calculated? What distribution shape is expected -there is a phrase referencing the normal dist, but is this reasonable given the skewed distribution of the scale to begin with?
How are the slopes assessed or compared? There are no statistics to support the similarity or difference of the log normal curves (e.g. lines 230-233).
Why compare adjacent items?

Validity of the findings

Statistical comparisons for generalizability of the curve fit and/or curve differences are needed.

Additional comments

I think the manuscript would benefit from summarizing the general trend noted here and then clarifying that it persists across items and clusters; it streamlines the results.
It could also be beneficial to emphasis the utility and value in the curve estimates or other curve/sub-score clusters

Reviewer 2 ·

Basic reporting

The article needs some minor editing, in general it is well written but it will need to be carefully edited - there are several instances of single sentence paragraphs, and run-on sentences.

Experimental design

The study is an analysis of archival data from a large data set. The analysis is appropriate, however, it's not entirely clear what the authors are trying accomplish with this study - nor is there any aprori hypothesis clear aside from they expect there to be a mathematical relationship.

Validity of the findings

The data analysis and model fitting are well done and valid, however, the authors need to take this paper to the next stop and clearly show how this result can be useful. The authors also need to test to determine if this is the best model - they describe a model that fits, but do not test alternative models to determine if theirs is the best fitting model. It may be that there is a better description of the data. They also have only described in qualitative terms how their data exhibit a mathematical pattern, they have not directly tested whether this pattern fits or not, merely described the pattern. Some 'goodness of fit' analysis would be appropriate here.

Additional comments

You have some interesting data and I think having the data available could be useful, however, you should quantify how closely your data resembles the mathematical functions and also determine if it is the best fit. Finally, spend some time explaining what the utility of this finding is.

---

## Round 0.2 · accepted · Accept

Thank you for your responses to the reviewers and addressing the comments. I believe it has created a stronger manuscript.